# The Frequency of Diseases within the Locomotor System Compared to Occupational Diseases of Salt Miners

Malwina Pietrzak *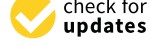 and Katarzyna Domaszewska *

Department of Physiology and Biochemistry, Poznan University of Physical Education,
Królowej Jadwigi Street 27/39, 61-871 Poznań, Poland
* Correspondence: m.pietrzak@awf.poznan.pl (M.P.); domaszewska@awf.poznan.pl (K.D.)

**Abstract:** The objective of this study was to determine the frequency of locomotor system diseases in salt miners compared with that of other occupational diseases. Methods: An analysis of diseases reported by salt miners working at different mining levels was carried out. All miners were asked about back pain in the past five years. The Oswestry Low Back Pain Disability Scale-Polish Version (OLBPDS-PL) and Neck Disability Index-Polish questionnaire Version (NDI-PL) were used to measure the functional disability of the lumbar and cervical spine. In contrast, the severity of low back pain was assessed using a 10 mm visual analog scale (VAS). In all, 62 miners were included in the study. Results: The most common diseases of salt miners are locomotor diseases involving the lumbar spine. The study showed a significant correlation between the occurrence of pain changes in the thoracic spine and the extraction level ($p < 0.05$). The extraction level also correlates with the reported level of pain in the thoracic spine ($p < 0.05$). The incidence of diseases such as hypertension and diabetes depended mainly on the age and weight of the subjects ($p < 0.05$). Hearing loss depended on the age of respondents and years of work in the mine ($p < 0.05$). Conclusions: Due to significant problems of miners in the field of the motor system, the list of occupational diseases in Poland, specified in the Regulation of the Council of Ministers of 30 June 2009 on occupational diseases (Journal of Laws No. 105, item 869) should be extended for example, to diseases affecting the spine. As there are no articles on the health of salt miners, further research should focus on complementing this knowledge to guide interventions to reduce the risk of chronic and occupational diseases. Extensive research is needed, including ergonomic measurements, to verify our results for the Polish salt mining industry.

**Keywords:** back pain; salt miner; occupational diseases; locomotor system; Neck Disability Index—NDI; Oswestry LBP questionnaire

## 1. Introduction

The mining industry, especially underground mining, is one of the most dangerous working environments. It is influenced by the occurrence of natural hazards and the complexity of the technological process (opening the deposit, its extraction, and transport) [1]. There is an inherent risk in the mining profession that is often impossible to predict. Physical effort, high temperature, humidity, high dustiness, and lower oxygen content in the mine atmosphere, often with the presence of harmful gases, are just a few of the problems that miners struggle with. The accompanying stress should not be forgotten, so the health condition of miners should be monitored in a special way [2]. The consequences of negligence are occupational diseases of miners, such as pneumoconiosis, permanent hearing loss, vibration syndrome, chronic bronchitis, and others [3]. In the entire mining industry, according to the data of the Polish Institute of Occupational Medicine, the National Research Institute, in 2015–2019, a total of 1656 cases of occupational diseases were found, of which pneumoconiosis had the largest share (1464 cases, i.e., 88.4% of all occupational diseases in mining). Permanent hearing loss was diagnosed in 103 miners in this period, which accounts for 6.21% of all occupational diseases in mining, while the other category

included 63 miners, which constitutes 3.8% [3]. The Regulation of the Council of Ministers of 30 June 2009 on occupational diseases (Journal of Laws No. 105, item 869) in force in Poland covers 26 occupational diseases. Unfortunately, among the chronic diseases of the locomotor system caused by the way work is performed, there is only chronic inflammation of the tendon and its sheath, chronic bursitis, chronic meniscal damage in people who work in a kneeling or squatting position, chronic periarticular inflammation of the shoulder, chronic epicondylitis of the humerus and fatigue bone fracture. Permanent muscle damage, diseases affecting the spine, or chronic arthritis (with the exception of the shoulder joint) are not included in the list [4].

Although the structure of the spine allows it to act as a support for the whole body, maintain balance, absorb shocks and protect the spinal cord and spinal nerves, overloads and injuries cause wear and tear on individual parts of the spine. At the same time, we cannot forget about bad habits and non-compliance with the principles of work ergonomics, which consequently lead to numerous dysfunctions [5]. Over time, chronic ailments appear that can expand their scope, causing pain and dysfunction in adjacent body segments that limit participation in professional and social life [6]. Heavy physical work related to a forced body position (static loads), lifting and carrying heavy objects (dynamic loads), and exposure to mechanical vibrations cause painful back ailments belonging to the category of work-related diseases [7–9].

Apart from excessive physical effort, the risk factors for diseases of the musculoskeletal system include the type of work performed, overweight, and obesity [10]. Numerous studies show that the incidence of back pain in miners is 69–78% [11–13]. However, this study is the first study of low back pain and other diseases in salt miners. It is also the first study that indicates the need to modify the list of occupational diseases in Poland. In the literature can be found many articles on the health of miners; however, there is no connection between back pain and the level of physical activity. The aim of this study is to determine the frequency of locomotor system diseases in salt miners compared with that of other occupational diseases.

## 2. Materials and Methods

### 2.1. Participants

The participants of the study were 62 male salt miners aged 25 to 63 years.

The study population was divided into 3 groups. Group 1 comprised surface workers; group 2, underground workers (mining level > 750 m below the ground); group 3, underground workers (mining level ≤ 750 m below the ground), according to the structure of the salt mine. The current mining level is 750–780 m below the surface, and the level up to 750 m below the ground is no longer exploited. The study used a proprietary questionnaire covering age, work experience, mining level, miners' diseases, back pain, and smoking. Among the miners surveyed, 13 (20.97%) people were current cigarette smokers.

The study was carried out at the NZOZ Rehabilitation, and Treatment Center in Kłodawa between February 24 and March 6 of 2020 from 6–8 am. They did not affect the work system in the mine. The study was conducted according to the Declaration of Helsinki and the National Statement of Intent as well as the Human Research Ethics Guidelines and was then approved by the IRB (Institute for Research in Biomedicine) at the Poznan University of Medical Sciences (19 June 2019; Ethics Approval Number: 695/19). Each recruited subject gave their written permission to take part after familiarizing themselves with the study protocol.

### 2.2. Anthropometric Measurements

The first anthropometric tests were performed each day. Participants were to refrain from drinking (except water) and eating for 3 h prior to the study.

Body height was measured using a Tanita HR-001 stadiometer (Tanita Corporation, Tokyo, Japan). The measurement range was 0–2.07 m (0–81-1/2 inches), and the graduation was 1 mm (1/8 inch).

Body mass was measured with WPT 60/150 OW medical scales with an accuracy of 100 g (Ragwag®, Radom, Poland). Differences between the three body weight and height measurements were observed to be <1%. The BMI is used to determine the degree of overweight and obesity in adults. This indicator is defined as the quotient of the body weight in kilograms, and the square of body height expressed in square meters.

### 2.3. Examinations of Back Pain Symptoms

After a light breakfast (one sandwich with butter and cheese, approximately 200 kcal), back pain surveys were conducted individually in the physiotherapy office. They took place in the presence of a physiotherapist who explained the participants. Examinations were carried out in the morning before the participants went to work.

#### 2.3.1. Oswestry LBP Questionnaire

In order to assess the quality of life of miners due to back pain in the lumbar spine, the Polish version of the scale of back pain and subsequent disability was used (Oswestry Low Back Pain Disability Scale-Polish Version, OLBPDS-PL). The scale includes 10 questions concerning pain intensity, lifting, sitting, sleeping, traveling, caring for, walking, standing, socializing, and changing pain intensity. Each question has six answers, each of which is scored from 0 to 5. The respondent's degree of disability is assessed on a scale of 0 to 50 points. The obtained values allowed for inclusion in one of the five disability groups. A score from 0 to 4 points characterizes the group of people in which disability is absent or minimal. A result from 5 to 14 points indicates mild disability; 15 to 24 points, moderate disability; 25 to 34 points, disability; and 35 to 50 points, extreme suffering and disability. The questionnaire is validated [14].

#### 2.3.2. Neck Disability Index-NDI

The quality of life of miners was assessed on the basis of the Polish version of the Neck Disability Index (Neck Disability Index-Polish questionnaire Version, NDI-PL). This questionnaire consists of 10 questions concerning pain intensity, care, picking up objects, reading, headaches, being able to concentrate, work, drive, sleep and rest. Each question has six answers, which are scored from 0 to 5. A score of 0–50 points or a percentage from 0% to 100% is obtained. Depending on the values obtained, the respondent can be classified into one of five disability groups. A score of 0–4 points indicates no or a minimum level of disability; 5–14 points, mild disability; 15–24 points, moderate disability; 25–34 points, severe disability; and 35–50 points, extreme suffering, and disability The questionnaire is validated [15].

#### 2.3.3. Visual Analogue Scale—VAS

The Visual Analogue Scale (VAS) is a universal tool for subjective pain assessment. The patient determines their perceived pain on a scale from 0 to 10, from the absence of pain to the strongest and unimaginable pain [16].

### 2.4. Statistics

All data are presented as mean ± standard deviation. The incidence of diseases was given as the number of people and as a percentage. The normality of the distribution was tested with the Shapiro–Wilk test. The differences between the variables were tested using one-way analysis of variance (ANOVA) for independent groups. Tukey's post hoc test was performed to assess significant differences between employees working at different levels. The relationship between the variables was tested while using results were statistically analyzed using Dell Statistica data analysis software system (version 13, software.dell.com, Dell Inc., Round Rock, TX, USA).

### 3. Results

A total of 62 salt mine employees working on three mining levels participated in the study. Some of the miners reported more than one disease listed in Table 1. The most frequently reported ailment was low back pain.

**Table 1.** The occurrence of chronic diseases among salt miners.

| Diseases | Participants (n = 62) [n (%)] | | | Total |
|---|---|---|---|---|
| | Group 1 (n = 13) | Group 2 (n = 16) | Group 3 (n = 33) | |
| Diabetes | 1 (1.61%) | 1 (1.61%) | 1 (1.61%) | 3 |
| Hypertension | 2 (3.22%) | 5 (8.06%) | 3 (4.84%) | 10 |
| Severe Cardiovascular disease | 0 (0%) | 0 (0%) | 0 (0%) | 0 |
| Severe Respiratory system disease | 0 (0%) | 0 (0%) | 0 (0%) | 0 |
| Neck back pain | 2 (3.22%) | 5 (8.06%) | 11 (17.74%) | 18 |
| Middle back pain | 1 (1.61%) | 3 (4.84%) | 12 (19.35%) | 16 |
| Low back pain | 6 (9.68%) | 14 (22.58%) | 19 (30.65%) | 39 |
| Hearing loss | 0 (0%) | 1 (1.61%) | 1 (1.61%) | 2 |
| Alcoholism | 0 (0%) | 0 (0%) | 2 (3.22%) | 2 |

Group 1—surface workers; Group 2—underground workers (level mining above 750 m below the ground; Group 3—underground workers level mining below or at level 750 m below the ground.

The obtained results showed statistically insignificant differences in such parameters as age, body height, body weight, seniority, and disability in the cervical and lumbar spine between miners working at different extraction levels (Table 2).

**Table 2.** Somatic characteristics and intensification of pain changes in the tested body areas of salt miners.

| Selected Anthropometric Indicators and Questionnaire Results | Participants (n = 62) [n (%)] | | | ANOVA p-Value |
|---|---|---|---|---|
| | Group 1 (n = 13) | Group 2 (n = 16) | Group 3 (n = 33) | |
| Age (yr) | 43.54 ± 8.637 | 42.56 ± 7.071 | 41.01 ± 7.691 | 0.7305 |
| Body weight (kg) | 93.39 ± 13.034 | 82.12 ± 11.242 | 87.31 ± 14.092 | 0.0584 |
| Body height (cm) | 178.71 ± 6.775 | 175.03 ± 4.745 | 174.49 ± 6.184 | 0.0991 |
| BMI (kg/m$^2$) | 29.29 ± 4.185 | 26.86 ± 5.081 | 28.65 ± 4.148 | 0.2408 |
| Seniority (yr) | 12.9 ± 7.751 | 15,84 ± 9.999 | 15.61 ± 8.663 | 0.7732 |
| NDI (points) | 3.31 ± 2.689 | 3.12 ± 3.519 | 3.85 ± 3.961 | 0.7688 |
| Oswestry (points) | 4.54 ± 6.050 | 8.62 ± 4.660 | 6.24 ± 6.428 | 0.0531 |
| VAS Cervical (points) | 0.85 ± 2.154 | 1.62 ± 2.156 | 1.33 ± 1.708 | 0.2971 |
| VAS Thoracic (points) | 0.62 ± 2.219 | 0.75 ± 2.176 | 1.18 ± 1.722 | 0.1130 |
| VAS Lumbar (points) | 2.38 ± 2.931 | 4.44 ± 2.308 | 3.36 ± 3.151 | 0.1790 |

Data are presented as mean ± standard deviation; BMI—body mass index; yr—years; NDI—Neck Disability Index; Oswestry–Oswestry Low Back Pain Disability Scale—Polish Version; VAS—Visual Analogue Scale.

Statistical analysis showed no difference between the studied anthropometric features and the intensification of pain changes in the tested body areas depending on the level

of work. The highest Oswestry result and VAS Thoracic and VAS Lumbar are found in miners working at the level of up to 750 m, while VAS Thoracic in persons working below 750 m. In this case, the extraction level correlates with the intensity of perceived pain (VAS Thoracic) (R = 0.2636 *p* = 0.0384). A positive correlation was also shown between the work level of miners and the declared pain in the thoracic region (R = 0.2713 *p* = 0.0329). Other analyzed diseases such as, hypertension or diabetes depended mainly on body weight (R = 0.2548 *p* = 0.0455) and age (R = 0.4345 *p* = 0.0004). Hearing loss was also associated with age and the number of years of work (R = 0.2836 *p* = 0.0255). Cardiovascular disease is more dependent on age and body weight, while hearing is more dependent on age than on extraction level.

## 4. Discussion

In this study of salt miners, the frequency of locomotor diseases was examined in comparison with that of other occupational diseases. The level of pain and disability of miners caused by back pain was also determined. The number of reported complaints in the motor system was significantly higher than that of respiratory and circulatory system diseases. Physical work requiring the use of heavy tools, uncomfortable, forced positions, repetitive movements, lifting heavy objects above the shoulder line, often exceeding the statutory 50 kg, and prolonged sitting without back support were the main causes of neck and shoulder pain, rotator cuff syndrome and epicondylitis [17,18].

There are many studies on miners' disease and back pain, such as those by Zejda et al. [2,8,19,20], but none of them consider it in the context of an occupational disease. According to the current list of occupational diseases contained in the Journal of Laws of the Republic of Poland of 25 November 2013, item 1367 [4], we do not include any diseases related to the spine as occupational diseases, and it is dysfunctions within the spine that prevent miners from continuing their professional work to the greatest extent. Despite the continuous improvement of working conditions, miners still carry out many tasks manually using various types of tools, which overload the spine and joints. As a consequence, they experience general musculoskeletal pain, arthritis, and osteoarthritis [18].

We cannot forget about the vibration syndrome to which miners are exposed, often underdiagnosed due to their non-stationary work. Many of them use pneumatic hammers in their work, but this is not their main job, so it is ignored in the occupational risk assessment. The use of hand-held power tools causes disturbances in the peripheral circulation in muscles, peripheral nerves, joints, and even bones [21,22]. The consequence of this is numbness of the fingers and entire upper limbs as well as night pains. Occupational exposure to vibration and a forced body position leads to musculoskeletal disorders in the neck and shoulders. Taking into account the exposure to base torsion, frequent bending, lifting, and noise, we obtain an almost 5-fold increase in the prevalence index [7,21–24]. High dynamic and/or static loads lead to acute injuries in the area of the cervical spine, lumbar spine, upper limbs, and feet. Ignoring the first symptoms of pain and discomfort may lead to serious health consequences, including discopathy, chronic back pain syndrome, and even disability. Initially, employees experience reduced productivity at work, frequent absences, the risk of losing their job, and over time, the need to change jobs, if possible. As a result, the employee experiences a decline in quality of life, financial problems, and depression [25]. Musculoskeletal disorders represent a huge financial burden for both employers and the public health system. This carries with it the costs of sick leave, shortages of qualified employees, the need to train other people, the costs of downtime, and even pensions and compensation. In the Ohio Workers' Compensation System, between 1999 and 2004, the annual cost of treating MSDs was nearly USD three billion [26]. The most frequently mentioned diseases of occupational miners are pneumoconiosis. In the mining of hard coal and metal ores, they account for 79.73% of all recorded cases. Salt miners are not a separate group but are combined with other mining and activities supporting mining and quarrying. Pneumoconiosis in this huge subgroup constitutes only 20.26%. This is due to the improved working conditions and even the special environmental conditions

in the salt mine, which are used in spa treatments. In Eastern Europe, natural salt caves are used to alleviate the symptoms of respiratory diseases [27]. A stable air temperature, air humidity between moderate and high, the presence of aerosol elements (potassium, sodium, calcium, and magnesium), and the lack of air pollution are unique features of the caves' microclimate [28]. Treatment with halotherapy using NaCl salt in various forms is associated with the alleviation of respiratory diseases such as asthma, COPD, bronchitis, and cystic fibrosis [27]. An example of such a health resort is the 'Wieliczka' Salt Mine in Wieliczka [29]. It is assumed that inhalation of hyperosmolar ionic solutions not only improves lung function in respiratory diseases but also facilitates mucociliary cleansing [30,31]. As shown by the research of Shumate et al. (2017), each group of miners should be interpreted separately due to the specificity of the work and the microclimate in the mine. Very often, miners are thrown into one group despite different threats and diseases, which is a big mistake [20]. Hence, it seems essential to revise the existing list of occupational diseases, especially when it comes to salt miners. The introduction of good diagnostics of musculoskeletal diseases in the future may reduce their occurrence and thus improve public finances related to the rehabilitation and payment of sickness benefits and compensation to this group of employees.

This study replenishes the lack of data on the diseases of salt miners. They show back pain and dysfunctions of the miners' locomotor system depending on the mining level at which they work.

## 5. Conclusions

The main conclusion from the conducted research is the need to verify the current list of occupational diseases in Poland. It will be possible by eliminating diseases that no longer occur (due to, for example, better working conditions) as well as supplementing the list with diseases that were not included before. The literature contains only articles on noise and related hearing loss [32], mechanical vibrations and vibration disease [33], as well as dust and pollution affecting respiratory diseases [34] or the cardiovascular system [35]. However, this is research generally related to mining. There are no articles covering the health of salt miners and their locomotor diseases. The theses presented in the study should be verified among miners working in other mining sectors in Poland and Europe.

The list of occupational diseases in force in Poland in accordance with the Regulation of the Council of Ministers of 30 June 2009 on occupational diseases (Journal of Laws No. 105, item 869) should be extended to include specific disease entities. Certainly, more accurate results could be obtained using imaging diagnostics: magnetic resonance imaging (MRI), computed tomography (CT), and X-ray examination (X-ray). Thanks to them, in addition to the pain scale, we would see the structure of the spine. For periodic examinations of employees, especially those working at the lowest extractive level, where there is a pain in every section of the spine, it would be worth introducing at least an X-ray examination. This would make it possible to observe the progress of changes in the spine in the future.

*Limitation of the Study*

The research was carried out in one of the three active salt mines in Poland. Carrying out analyzes among miners of other salt mines would allow broadening the variability of the occurrence of diseases of salt miners across Poland.

**Author Contributions:** Conceptualization, K.D. and M.P.; Methodology, K.D.; Investigation M.P., Resources M.P., Data Curation M.P. and K.D., Writing—Original Draft Preparation M.P. and K.D., Visualization, M.P. and K.D, Supervision K.D., Project Administration K.D. All authors have read and agreed to the published version of the manuscript.

**Funding:** This research received no external funding.

**Informed Consent Statement:** Informed consent was obtained from all subjects involved in the study.

**Data Availability Statement:** The data presented in this study are available on request from the corresponding author. The data are not publicly available due to the consent provided by partici-pants on the use of confidential data.

**Conflicts of Interest:** The authors declare no conflict of interest.

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
