# Peer review of "The Frequency of Diseases within the Locomotor System Compared to Occupational Diseases of Salt Miners"

_sustainability, doi:10.3390/su14169857_

Round 1

Reviewer 1 Report

Dear Authors,

The paper presents a topic of interest for researchers and practitioners. However, several improvements are needed.

1. The abstract should present the need for research, the methodology, the main results obtained and the future directions of research.

2. The introduction should be supplemented with other research conducted in this field and with other developed models developed. In this way the research will show what is the gap it fills and what are the elements of originality.

3. To emphasize the need for this study.

4. The stages of the methodology must be presented in detail.

5. To highlight the gaps filled by the present study.

6. The conclusions section should be completed with a review of the study.

Author Response

In the beginning, we would like to thank you for the effort you have put into making our manuscript more precise and more preferable. We have made some significant changes according to Your precious suggestion. 

Our responses to the comment are described in manuscript. Added parts are in yellow color.

Best regards,

Authors

  1. The abstract should present the need for research, the methodology, the main results obtained and the future directions of research.

Reply In response to the suggestion, I supplemented the abstract with the required data.

Abstract

Objectives: The objective of this study was to determine the frequency study of locomotor system diseases in salt miners compared with that of other occupational diseases. Methods: An analysis of diseases reported by salt miners working at different mining levels was carried out. All miners were asked about back pain in the past 5 years. The Oswestry Low Back Pain Disability Scale - Polish Version (OLBPDS-PL) and Neck Disability Index - Polish questionnaire Version (NDI-PL) were used to measure the functional disability of the lumbar and cervical spine. In contrast, the severity of low back pain was assessed using a 10 mm visual analog scale (VAS). In all 62 miners were included in the study. Results: The most common diseases of salt miners are locomotor diseases involving the lumbar spine. The study showed a significant correlation of the occurrence of pain changes in the thoracic spine depending on the level at which salt is extracted (R = 0.2636 p = 0.0384).The incidence of diseases such as hypertension and diabetes depended mainly on the age and weight of the subjects (p < 0.05). Hearing loss depended on the age of respondents and years of work in the mine (p < 0.05). Conclusions: Due to significant problems of miners in the field of the motor system, the list of occupational diseases in the Regulation of the Council of Ministers of 30 June 2009 on occupational diseases (Journal of Laws No. 105, item 869) should be extended for example, to diseases affecting the spine. As there are no articles on the health of salt miners, further research should focus on complementing this knowledge to guide interventions to reduce the risk of chronic and occupational diseases. Extensive research is needed, including ergonomic measurements, to verify our results for the Polish salt mining industry.

  1. The introduction should be supplemented with other research conducted in this field and with other developed models developed. In this way the research will show what is the gap it fills and what are the elements of originality.

Reply Thank you for your suggestions. The following text was added to the introduction.

Apart from excessive physical effort, the risk factors for diseases of the musculoskeletal system include the type of work performed, overweight and obesity [10]. Numerous studies show that the incidence of back pain in miners is 69–78% [11-13]. However, this study is the first study of low back pain and other diseases in salt miners. It is also the first study that indicates the need to modify the list of occupational diseases in Poland. The aim of the study is to determine the frequency of locomotor system diseases in salt miners compared with that of other occupational diseases.

  1. To emphasize the need for this study.

Reply This study shows the need to re-analyze the current list of occupational diseases in Poland, which has not been modified since 2013. At that time, the conditions and specificity of miners' work changed in many mines.

  1. The stages of the methodology must be presented in detail.

Reply The questionnaires, disability and pain assessments were carried out by a qualified physiotherapist employed at the NZOZ Rehabilitation and Treatment Center in KÅ‚odawa. The study was carried out between February 24 and March 6 of 2020 from 6-8 am. They didn’t affect the work system in the mine.

2.2. Anthropometric Measurements

                Reply The first anthropometric tests were performed each day. Participants were to refrain from drinking (except water) and eating for 3 hours prior to the study.

2.2.3. Examinations of back pain symptoms

After a light breakfast (one sandwich with butter and cheese approximately 200 kcal), back pain surveys were conducted individually in the physiotherapy office. They were held in the presence of a physiotherapist who explained the participants. Examinations were carried out in the morning before the participants went to work.

2.2.3.1 Oswestry LBP Questionnaire

The questionnaire were carried out by a qualified physiotherapist employed at the NZOZ Rehabilitation and Treatment Center in KÅ‚odawa. The questionnaire is validated [11]

  1. To highlight the gaps filled by the present study.

Reply This study replenishes the lack of data on diseases of salt miners. They show back pain and dysfunctions of the miners' locomotor system depending on the mining level at which they work.

  1. The conclusions section should be completed with a review of the study.

Reply The main conclusion from the conducted research is the need to verify the current list of occupational diseases in Poland. It will be possible by eliminating diseases that no longer occur (due to, for example, better working conditions) as well as supplementing the list with diseases that were not included before. The literature contains only articles on noise and the related hearing loss [30], mechanical vibrations and vibration disease [31], as well as dust and pollution affecting respiratory diseases [32] or the cardiovascular system [33]. However, this is research generally related to mining. There are no articles covering the health of salt miners and their locomotor diseases. The theses presented in the study should be verified among miners working in other mining sectors in Poland and Europe.

Reviewer 2 Report

This paper deals with an analysis on locomotor disease frequency in saltminers. The paper is fair work, is well structured, and presents interesting results. The topic seems to be underaddressed in recent years, so this paper is relevant. A few concerns remain.

General spelling and English revisions are necessary.

Formatting / template issue: the running title appears appended to the paper title, instead of as a running title.

In the abstract, the objective begins stated as "to determine the frequency study of locomotor system diseases" - it appears the word "study" does not belong. Also in the abstract, why are p values presented in two ways? (as value and as upper limit).

In the conclusions and in the abstract, the country of the cited legislation should be mentioned.

Introduction: The country of the organization mentioned on line 37 should be mentioned. At line 64, suggest replacing "The aim of the study" with "The aim of this study" to avoid confusion as to this last sentence referring to reference [10].

Methods: Sect.2.1. The grouping of participants related to mining levels requires clarification. How is group 2 formed if the mine does not exploit above 750m?

Sect.2.2. Sensors / measurement tools, if possible (where available), should be followed by a reference to the manufacturer website or datasheet.

Sect.2.4-2.6. A brief mention if the scales are validated could round up the sections nicely.

Results: Line 137, table 1 - is the intent behind "low back pain" to mean not a lot of back pain? Or pain of the lower back (in which case it should be changed to "lower back pain")? Table 1: the addition of "(number of people)" on each line of column 1 seems unnecessary since the units are given on the first row as "[n (%)]". Table 1: why is diabetes in bold? Table 2: why is age in bold?

Tables: the description of table items should go into the caption, instead of under the table. It becomes difficult to read when mixed with the rest of the text.

Line 158: "body weight and weight" there seems to be a mistake here? Was the second weight meant to be height?

Discussion: There is no discussion on the limitations of the study. Is the number of participants significant to the profession in Poland? Did the morning examination affect the schedule of participants in any way that could affect collected data? Was the examiner the same person every time? Any other potential source of bias in the study?

Author Response

In the beginning, we would like to thank you for the effort you have put into making our manuscript more precise and more preferable. We have made some significant changes according to Your precious suggestion. 

Our responses to the comment are described in manuscript. Added parts are in yellow color.

Best regards,

Authors

1.General spelling and English revisions are necessary.

Reply Thank you for your attention while writing the manuscript, the authors used the linguistic proofreading done by Proof-reading-service.com.

2.Formatting / template issue: the running title appears appended to the paper title, instead of as a running title.

Reply All comments were included in the text of the manuscript.

3.In the abstract, the objective begins stated as "to determine the frequency study of locomotor system diseases" - it appears the word "study" does not belong. Also in the abstract, why are p values presented in two ways? (as value and as upper limit).

Reply Thank you for your attention, I deleted the word "study" and unified all values to the p< 0.05 system

4.In the conclusions and in the abstract, the country of the cited legislation should be mentioned.

Reply Thanks for your suggestion. Both the conclusions and the abstract were supplemented with the country of the cited legislation.

5.Introduction: The country of the organization mentioned on line 37 should be mentioned. At line 64, suggest replacing "The aim of the study" with "The aim of this study" to avoid confusion as to this last sentence referring to reference [10].

Reply Thank you for your attention. I completed the country of the organization mentioned on line 37, as well as at line 64 I replaced "The aim of the study" with „The aim of this study”.

6.Methods: Sect.2.1. The grouping of participants related to mining levels requires clarification. How is group 2 formed if the mine does not exploit above 750m?

Reply In the salt mine covered by the research, the levels of exploitation are at the level of 750 m as well as at the depth of 810 m. The first group consists of surface workers who are also the control group. The second group consists of miners working up to a depth of 750 m below the ground (electricians, mechanics, supervision workers), while the third group consists of workers working below 750 m below the surface (face miners, transport workers).

7.Sect.2.2. Sensors / measurement tools, if possible (where available), should be followed by a reference to the manufacturer website or datasheet.

Reply Thank you for your attention. In the description of the measurement tools (Oswestry LBP Questionnaire, Neck Disability Index - NDI, Visual Analogue Scale - VAS), there are references in the text:

[14]Misterska, E.; Jankowski, R.; GÅ‚owacki, M. Quebec Back Pain Disability Scale, Low Back Outcome Score and Revised Oswestry Low Back Pain Disability Scale for Patients with Low Back Pain Due to Degenerative Disc Disease. Spine. 2011, 36, E1722-E1729; DOI:10.1097/BRS.0b013e318216ad48.

[15]Misterska, E.; Jankowski, R.; GÅ‚owacki, M. Cross-cultural adaptation of the Neck Disability Index and Copenhagen Neck Functional Disability Scale for patients with neck pain due to degenerative and discopathic disorders. Psychometric properties of the Polish versions. BMC Musculoscelet Disord. 2011, 84, 2474-2784; DOI:10.1186/1471-2474-12-84.

[16] Hawker, G.A; Mian, S.; Kendzerska, T.; French, M. Measures of adult pain: Visual Analog Scale for Pain (VAS Pain), Numeric Rating Scale for Pain (NRS Pain), McGill Pain Questionnaire (MPQ), Short-Form McGill Pain Questionnaire (SF-MPQ), Chronic Pain Grade Scale (CPGS), Short Form-36 Bodily Pain Scale (SF-36 BPS), and Measure of Intermittent and Constant Osteoarthritis Pain (ICOAP). American College of Rheumatology. 2011, 63, S240 –S252; DOI:10.1002/acr.20543.

  1. Sect.2.4-2.6. A brief mention if the scales are validated could round up the sections nicely.

Reply Thank you for your attention, the sections have been rounded and the tests used are validated.

9. Results: Line 137, table 1 - is the intent behind "low back pain" to mean not a lot of back pain? Or pain of the lower back (in which case it should be changed to "lower back pain")?

Reply While preparing the article, I was guided by the nomenclature used in other studies. In the literature, lower back pain is referred to as low back pain:

  1. Sarikaya S, Ozdolap S, GümüÅŸtasÅŸ S, Koç U. Low back pain and lumbar angles in Turkish coal miners. Am J Ind Med. 2007 Feb;50(2):92-6. doi: 10.1002/ajim.20417. PMID: 17238134.
  2. NakipoÄŸlu GF, Karagöz A, Ozgirgin N. The biomechanics of the lumbosacral region in acute and chronic low back pain patients. Pain Physician. 2008 Jul-Aug;11(4):505-11. PMID: 18690279.
  3. Bozorgmehr A, Ebrahimi Takamjani I, Akbari M, Salehi R, Mohsenifar H, Rasouli O. Effect of Posterior Pelvic Tilt Taping on Abdominal Muscle Thickness and Lumbar Lordosis in Individuals With Chronic Low Back Pain and Hyperlordosis: A Single-Group, Repeated-Measures Trial. J Chiropr Med. 2020
  4. Table 1: the addition of "(number of people)" on each line of column 1 seems unnecessary since the units are given on the first row as "[n (%)]". Table 1: why is diabetes in bold? Table 2: why is age in bold? Tables: the description of table items should go into the caption, instead of under the table. It becomes difficult to read when mixed with the rest of the text.

Reply All comments have been included in the main body of the manuscript.

11.Line 158: "body weight and weight" there seems to be a mistake here? Was the second weight meant to be height?

Reply Thank you for your attention word body weight has been corrected for age. It was an editorial error.

12.Discussion: There is no discussion on the limitations of the study. Is the number of participants significant to the profession in Poland

Reply The limitation chapter has been added to the main manuscript.

The research was carried out in one of the three active salt mines in Poland. Carrying out analyzes among miners of other salt mines would allow to broaden the variability of the occurrence of diseases of salt miners across Poland.

  1. Did the morning examination affect the schedule of participants in any way that could affect collected data? Was the examiner the same person every time? Any other potential source of bias in the study?

Reply The tests were always performed by 1 physiotherapist between 6-8 am in the NZOZ Rehabilitation and Treatment Center in KÅ‚odawa. The results of the research were used only for scientific work and did not cause any reorganization of work in the mine.

Reviewer 3 Report

The paper with the running title “Occupational diseases of salt miners” addresses whether salt miners are developing musculoskeletal disorders of the spine.  The authors point out that currently, Polish regulations do not recognize spinal musculoskeletal disorders as an occupational disease.  This paper intends to provide evidence of the occupational origins by surveying Polish salt miners about their neck and low back and spinal pain.

I commend the authors for pursuing this worthy question.  However, the evidence they present doesn’t support their conclusion that “the list of occupational diseases in the Regulation of the Council of Ministers of 30 June 2009 on occupational diseases….should be extended for example, to diseases affecting the spine”.

The evidence of disease is not compelling because

1.      No significant differences in any of the surveys was found between the three groups except for body weight where Group 1 appears to be heavier. 

2.      The groups seem to be based on the depth at which the participants work (surface, < 750 meters, > 750 meters) but there is no additional description of whether they engage in activities having different levels of risk for musculoskeletal injury (repeated motion, lifting, carrying, etc).

3.      There’s no control group (ideally it would comparable men working in other occupations) so we really don’t know whether the levels of pain reported are related to their work as salt miners or just typical for men with their characteristics. 

4.      The results were analyzed mainly by ANOVA.  A multivariate analysis could have investigated the role of work (tasks/group, length of employment) as variables along with age, weight, and other non-occupational factors. Doing so could help tease whether occupation plays a role in the observed effects.

In addition, the usual practice is to present univariate statistics describing the workers – age, duration of employment, education -- separately from the outcome variables.

Other comments:

·        Line 15-16 The text says “The study showed a significant correlation of the occurrence of pain changes in the thoracic spine depending on the extraction level (R = 0.2636 p = o.0384)”.  What do you mean by extraction level? Are you talking about the three groups you defined?  Secondly, the data presented in Table 2 summarizes thoracic pain and from what I can see the results were not significant.  Where does the p = 0.0384 come from?

·        Table 1 – how were the diseases ascertained?  I find it surprising that there are 0 cases of either cardiovascular disease or respiratory system disease among 62 male workers whose average age is in their 40’s. 

·        Table 2 – Do the numbers in the column under “ANOVA” correspond to p values?

·        Lines 156-157 The statistic from the abstract is shown here so please clarify what is being described.

·        Lines 157-158 You say that diabetes, hypertension and hearing loss depended on “body weight and weight”.  How was this determined?  Did you look at the relationship between the pain variables and body weight, seniority etc? 

·        Lines 159-160 says that hearing loss was associated with age and number of years worked with the results reported in Table 2.  Table 2 shows that ANOVA results.  Is there a missing Table?

Author Response

In the beginning, we would like to thank you for the effort you have put into making our manuscript more precise and more preferable. We have made some significant changes according to Your precious suggestion. 

Our responses to the comment are described in manuscript. Added parts are in yellow color.

Best regards,

Authors

  1. No significant differences in any of the surveys was found between the three groups except for body weight where Group 1 appears to be heavier. 

Reply Thank you for your valuable comment. The conclusion from my research is as follows: musculoskeletal diseases do not depend on the mining level at which miners work, but they are most common among salt miners and this applies to all mining levels. The groups differed in terms of body weight, but this difference was not statistically significant p = 0.058. A positive correlation A positive correlation was demonstrated between the declared back pain in the thoracic section and the extraction level (R = 0.2713 p = 0.0329), as well as a positive correlation between the analogy VAS scale and the extraction level at which salt miners work (R = 0.2636 p = 0.0384).

  1. The groups seem to be based on the depth at which the participants work (surface, < 750 meters, > 750 meters) but there is no additional description of whether they engage in activities having different levels of risk for musculoskeletal injury (repeated motion, lifting, carrying, etc).

Reply Thank You for your valuable comment . The groups are divided according to the mining level at which they work, but the activities and most of the movements they perform are similar. At all levels, miners carry, lift, bend, participate in transportation, operate machinery, stand or walk for long periods of time. Only the conditions under which they perform their work change. From the point of biomechanical stress, miners in group III have the greatest risk of stressing the locomotor system.

  1. There’s no control group (ideally it would comparable men working in other occupations) so we really don’t know whether the levels of pain reported are related to their work as salt miners or just typical for men with their characteristics. 

Reply Thank You for your valuable comment. The control group for the study consisted of surface miners. Miners cannot be compared to other professions because they have specific workloads to the locomotor system and work specificity, and the research was to concern salt miners and the list of occupational diseases in Poland, where spine diseases were omitted. Spine pain in the population affects both women and men, but the main goal of my work was to show the prevalence of back pain in the professional group of miners. The list of occupational diseases for them is limited to pneumoconiosis, hearing loss, and as my research shows, salt miners have the greatest problems with the locomotor system, therefore changes in the current regulation are important.

  1. The results were analyzed mainly by ANOVA.  A multivariate analysis could have investigated the role of work (tasks/group, length of employment) as variables along with age, weight, and other non-occupational factors. Doing so could help tease whether occupation plays a role in the observed effects.

In addition, the usual practice is to present univariate statistics describing the workers – age, duration of employment, education -- separately from the outcome variables.

Reply Thank You for your valuable comment. In the study, the results were analyzed using ANOVA as an effective method to demonstrate the significance of differences between the study groups. Additionally, the correlations between the studied variables were also calculated.

Other comments:

  • Line 15-16 The text says “The study showed a significant correlation of the occurrence of pain changes in the thoracic spine depending on the extractionlevel (R = 0.2636 p = o.0384)”.  What do you mean by extraction level? Are you talking about the three groups you defined?  Secondly, the data presented in Table 2 summarizes thoracic pain and from what I can see the results were not significant.  Where does the p = 0.0384 come from?

Reply Thank You for your valuable comment. Extraction level - is one of the three separate mining levels in the salt mine.My study includes three levels (groups defined by me):Group 1 – surface workers; Group 2 – underground workers (level mining above 750 m below the ground; Group 3 – underground workers level mining below or at level 750 m below the ground. With a small group of miners, the subjective assessment of pain intensity (VAS) showed that the deeper the miners work, the more pain they feel in the thoracic spine, and it was a statistically significant difference p < 0.05. However, with the increase in the size of the group, this correlation did not work out. This fragment has been removed from the work.

  • Table 1 – how were the diseases ascertained?  I find it surprising that there are 0 cases of either cardiovascular disease or respiratory system disease among 62 male workers whose average age is in their 40’s. 

        Reply Thank You for your valuable comment. The diseases were reported by miners participating in the study and concerned only severe respiratory diseases (cancer, COPD, pneumoconiosis, tuberculosis) and severe cardiovascular diseases (heart attack, ischemic heart disease, valvular heart disease, coronary heart disease, stroke), hypertension is described separately. Table no. 1 has been completed.

  • Table 2 – Do the numbers in the column under “ANOVA” correspond to p values?

        Reply Thank You for your valuable comment. Column under “ANOVA” correspond to p values. This has been completed in the table.

  • Lines 156-157 The statistic from the abstract is shown here so please clarify what is being described.

Reply Thank You for your valuable comment. In the case, the extraction level correlates with the intensity of perceived pain (VAS Thoracic) (R = 0.2636 p = 0.0384). A positive correlation was also shown between the work level of miners and the declared pain in the thoracic region (R = 0.2713 p = 0.0329). Other analyzed diseases such as, hypertension or diabetes depended mainly on body weight (R = 0.2548 p = 0.0455) and age (R = 0.4345 p = 0.0004). Hearing loss was also associated with age and the number of years of work (R = 0.2836 p = 0.0255). Cardiovascular disease is more dependent on age and body weight, while hearing is more dependent on age than on extraction level.

  • Lines 157-158 You say that diabetes, hypertension and hearing loss depended on “body weight and weight”.  How was this determined?  Did you look at the relationship between the pain variables and body weight, seniority etc? 

Reply Thank You for your valuable comment. There was no correlation between the variables describing pain and seniority, body weight, body hight, etc. Diabetes and hypertension depended on “body weight and age” Editorial error, this has been corrected in the text.

  • Lines 159-160 says that hearing loss was associated with age and number of years worked with the results reported in Table 2.  Table 2 shows that ANOVA results.  Is there a missing Table?

Reply Thank You for your valuable comment. Table 2 was incorrectly placed because the correlations were described and their results were not included in the table. This has been corrected.

Round 2

Reviewer 1 Report

Dear Authors,

I accept the present form of the paper.

Best regards,

Reviewer

Reviewer 3 Report

Thank you for your revisions.